# Identification and Characterization of an OSH1 Thiol Reductase from *Populus Trichocarpa*

**DOI:** 10.3390/cells9010076

**Published:** 2019-12-27

**Authors:** Hui Wei, Jie Zhou, Chen Xu, Ali Movahedi, Weibo Sun, Dawei Li, Qiang Zhuge

**Affiliations:** 1Co-Innovation Center for Sustainable Forestry in Southern China, Key Laboratory of Forest Genetics & Biotechnology, Ministry of Education, College of Biology and the Environment, Nanjing Forestry University, Nanjing 210037, China; 15850682752@163.com (H.W.); xuchenidea@hotmail.com (C.X.); ali_movahedi@njfu.edu.cn (A.M.); czswb@njfu.edu.cn (W.S.); dwli@njfu.edu.cn (D.L.); 2Jiangsu Academy of Forestry, Nanjing 211153, China; zjwin718@126.com; 3Jiangsu Provincial Key Construction Laboratory of Special Biomass Resource Utilization, Nanjing Key Laboratory of Quality and Safety of Agricultural Products, Nanjing Xiaozhuang University, Nanjing 211171, China

**Keywords:** cadmium, disulfide bond, GILT, *Populus trichocarpa*, PtOSH1, ROS-scavenging system

## Abstract

Interferon gamma-induced lysosomal thiol reductase (GILT) is abundantly expressed in antigen-presenting cells and participates in the treatment and presentation of antigens by major histocompatibility complex II. Also, GILT catalyzes the reduction of disulfide bonds, which plays an important role in cellular immunity. (1) Background: At present, the studies of GILT have mainly focused on animals. In plants, GILT homologous gene (*Arabidopsis thaliana OSH1*: *AtOSH1*) was discovered in the forward screen of mutants with compromised responses to sulphur nutrition. However, the complete properties and functions of poplar OSH1 are unclear. In addition, CdCl_2_ stress is swiftly engulfing the limited land resources on which humans depend, restricting agricultural production. (2) Methods: A prokaryotic expression system was used to produce recombinant PtOSH1 protein, and Western blotting was performed to identify its activity. In addition, a simplified version of the floral-dip method was used to transform *A. thaliana*. (3) Results: Here, we describe the identification and characterization of OSH1 from *Populus trichocarpa*. The deduced PtOSH1 sequence contained CQHGX2ECX2NX4C and CXXC motifs. The transcript level of *PtOSH1* was increased by cadmium (Cd) treatment. In addition, recombinant PtOSH1 reduced disulfide bonds. A stress assay showed that *PtOSH1*-overexpressing (OE) *A. thaliana* lines had greater resistance to Cd than wild-type (WT) plants. Also, the activities of superoxide dismutase (SOD), peroxidase (POD), and catalase (CAT) in *PtOSH1*-OE plants were significantly higher than those in WT *A. thaliana*. These results indicate that PtOSH1 likely plays an important role in the response to Cd by regulating the reactive oxygen species (ROS)-scavenging system. (4) Conclusions: PtOSH1 catalyzes the reduction of disulfide bonds and behaves as a sulfhydryl reductase under acidic conditions. The overexpression of *PtOSH1* in *A. thaliana* promoted root development, fresh weight, and dry weight; upregulated the expression levels of ROS scavenging-related genes; and improved the activity of antioxidant enzymes, enhancing plant tolerance to cadmium (Cd) stress. This study aimed to provide guidance that will facilitate future studies of the function of PtOSH1 in the response of plants to Cd stress.

## 1. Introduction

Interferon-γ-inducible lysosomal thiol reductase (GILT) has been discovered in many species [1,2]. The precursor of GILT is a soluble glycoprotein, which enters the lysosome by interacting with the mannose-6-phosphate receptor and is transformed into the mature form by cleavage at the N- and C-termini [3,4]. In the immune system, GILT participates in the denaturation, renaturation, and reduction of disulfide bonds, which are essential for antigen processing and presentation [2,5]. Also, GILT enhances the efficiency of tyrosinase-related protein 1 (TRP-1) presentation by antigen-presenting cells, thus promoting the activity of T cells and autoimmunity [6]. In addition, GILT participates in the processing of human leukocyte antigen class II, which is essential for activating CD4+ T cells [7,8]. Moreover, GILT enhances the activity of superoxide dismutase (SOD) 2 and maintains reactive oxygen species (ROS) homeostasis [6,9]. GILT has been studied in animals, but its function in plants is unclear.

As the main source of inorganic pollution in soil and farmland [10,11], cadmium (Cd) affects plant growth and development, membrane permeability, photosynthesis, antioxidant enzymes, macromolecular structure, and metabolism, eventually leading to death [12,13]. Cd stress induces ROS production in plants, the accumulation of which hinders photosynthesis and compromises the metalloproteins related to electron transfer, suppressing respiration [14]. Cd toxicity can lead to oxidative stress and protein denaturation, to which plants respond by increasing their ability to remove oxidized proteins, enhancing their antioxidant capacity and synthesis of molecular chaperones, and changing the composition of the cell wall and xylem [15,16,17,18]. Superoxidase (SOD), catalase (CAT), peroxidase (POD), and other antioxidant enzymes in plants constitute a ROS-scavenging system, which reduces damage to membranes and facilitates adaptation to adverse environments. In plants, SOD degrades the superoxide anion into H_2_O_2_ and O_2_, which represents the first defense against ROS; also, POD and CAT oxidize H_2_O_2_, further alleviating oxidative damage [19,20,21].

Thioredoxin (Trx), which has a similar structure as GILT, has been widely studied [22,23,24,25,26,27]. Trx has a typical WCGH/PCK active domain, in which two cysteine residues are highly conserved and participate in the catalytic reduction of disulfide bonds of target proteins and the regulation of enzyme activity. The molecular structure of the WCGH/PCK domain is similar to that of CXXC, the active site of GILT, which can catalyze the reduction of disulfide bonds [22]. In redox reactions, Trx in its reduced state contacts the disulfide bond of the target protein through the hydrogen on the sulfhydryl, and transforms its own sulfhydryl into a disulfide bond. At the same time, the disulfide bond on the target protein is disrupted, completing the reduction of Trx [28,29]; GILT-mediated reduction occurs in a similar manner. The biological activity of Trx depends on the redox power of reversible disulfide bonds, which can affect protein activity and homeostasis. Disulfide bonds can stabilize protein conformation, which is important for maintaining the spatial organization of a protein. Trx oxidation-reduction is involved in various physiological and biochemical reactions, including redox balance, antioxidant effects [30], signal transduction [31,32], regulation of transcription factors [33,34], and response to heavy metal stress [35]. Trx can function as an antioxidant to remove ROS or as a repair enzyme to regenerate proteins inactivated by oxidative stress. Treating *Chlamydomonas reinhardtii* with 100 μm CdCl_2_ caused to increase the expression of Trx [36] and its activity by adding dithiothreitol (DTT). The expression of Trx is not able to activate nicotinamide adenine dinucleotide phosphate-dependent malate dehydrogenase (NADP-MDH). It has been suggested that the Trx heavy metal binding site is related to the WCGH/PCK active domain. In addition, the *OSH1* gene (At5g01580) improved transcript levels of endogenous S-responsive genes and led to O-Acetyl-L-Ser concentration in the rosette leaves [37]. These studies provide a new theoretical basis for the characterization of OSH1 from plants.

Poplar is an important tree as a source of industrial raw material and is widely distributed in temperate and cold temperate regions. The availability of its genome sequence [38] has promoted genetic research on poplar. However, no in-depth analysis of the *OSH1*gene of *Populus trichocarpa*, the putative product of which catalyzes the reduction of disulfide bonds, has been conducted. Furthermore, the role of OSH1 in resistance to Cd stress is unclear. In this study, we isolated the *PtOSH1* (POPTR_0016s12980) gene from *P. trichocarpa*. Recombinant PtOSH1 catalyzed the reduction of disulfide bonds. Additionally, the expression of *PtOSH1* was high in leaves and roots and responded to Cd. The enhanced resistance to Cd caused by *PtOSH1* may be mediated by the reparation of antioxidant enzymes and promotion of ROS homeostasis.

## 2. Materials and Methods

### 2.1. Plant Cultivation, Gene Isolation, and Vector Construction

Poplar (*Populus trichocarpa*) and *Arabidopsis thaliana* were cultivated in half-strength Murashige and Skoog (1/2 MS) medium (23 °C, humidity 74%, and 16 h light/8 h darkness). The young leaves, mature leaves, stems, roots, and petioles of *P. trichocarpa* were collected (Appendix A) and total RNA from different tissues was extracted based on the manufacturer’s instructions (Biomiga, San Diego, CA, USA). In addition, *P. trichocarpa* seedlings treated with 200 μM ABA, 2 mM CdCl_2_, and 2 mM H_2_O_2_ and *P. trichocarpa* young leaves treated with the different abiotic stresses were collected at 0, 2, 4, 6, 8, 12, 24, and 48 h. Forward and reverse primers were designed based on the sequence of *P. trichocarpa* GILT obtained from the National Center for Biotechnology Information, and *PtOSH1* was amplified by polymerase chain reaction (PCR). The PCR product was inserted into the PEASY-T3 vector (TransGen Biotech, Beijing, China), which was sequenced by Nanjing Genscript Co. (Nanjing, China). The PCR product with *EcoR*I and *Xho*I restriction sites was inserted into the PET-28a vector and the PCR product with *Xba*I and *BamH*I restriction sites was inserted into the PBI121 vector by double digestion and T4 ligation. The primers used are shown in the Appendix A.

### 2.2. Prokaryotic Expression, Purification, and Activity of PtOSH1 In Vitro

The recombinant plasmid PET-28a-*PtOSH1* was transformed into *Escherichia coli* BL21 (DE3) and cultured in Luria–Bertani (LB) medium. The optical density of *E. coli* BL21 (DE3) at 600 nm was determined and the isopropyl β-D-1-thiogalactopyranoside (IPTG) was added to induce synthesis of recombinant PtOSH1. We evaluated the effect of temperature (37 and 16 °C) and rotational speed (220 and 110 rpm) on PtOSH1 synthesis. The PtOSH1 level was determined by 12% sodium dodecyl sulfate-polyacrylamide gel electrophoresis (SDS-PAGE). A nickel-nitrilotriacetic acid (Ni-NTA)-chelating column was used to purify PtOSH1. The supernatant of recombinant *E. coli* BL21 (DE3) cultures was added to the column, followed by wash buffer (20 mM Tris-HCl, 0.15 M NaCl, and 20 mM imidazole; pH 8.0) until the absorbance reached baseline, and finally elution buffer (20 mM Tris-HCl, 0.15 M NaCl, and 250 mM imidazole; pH 8.0).

To assay PtOSH1 activity, 0.2% SDS was used to denature immunoglobulin G (IgG) at 100 °C for 5 min. Dilution buffer (50 mM NaCl, 0.1% Triton X-10) was used to dilute the denatured IgG. PtOSH1 was dissolved in enzyme activity solution (100 mM NaCl, 0.1% Triton X-10, 50 mM CH_3_COONa, 25 mM DTT; pH 4.5), and the mixture was incubated at 37 °C for 10 min to pre-activate PtOSH1. Also, 10 µL of denatured IgG and 100 µL of PtOSH1 were incubated at 37 °C for 1 h and subjected to non-reducing SDS-PAGE.

### 2.3. Overexpression of PtOSH1 in Arabidopsis Thaliana (A. Thaliana)

The recombinant plasmid PBI121-*PtOSH1* was transformed into *Agrobacterium tumefaciens* GV3101, and subsequently transformed into *A. thaliana* by the floral-dip method [39]. Putative T1 seedlings were screened in 1/2 MS medium containing 50 µg mL^−1^ kanamycin, and the putative plants were grown in soil. Subsequently, the T2 seedlings were screened as above. The DNA and RNA of T2 plants were extracted to confirm the insertion of *PtOSH1* into the *A. thaliana* genome. Three independent biological experiments were performed.

For CdCl_2_ treatment, wild-type (WT) and *PtOSH1*-OE *A. thaliana* plants were cultivated in 1/2 MS medium, and 1-week-old WT and *PtOSH1*-OE *A. thaliana* plants were planted in 1/2 MS medium with 0, 20, 40, 60, and 80 μM CdCl_2_. WT and *PtOSH1*-OE *A. thaliana* plants were grown in MS medium containing 0–80 μM CdCl_2_ for 20 days, and the root length, fresh weight, and dry weight of WT and *PtOSH1*-OEs were determined. Three independent biological experiments were performed. In addition, the transcript levels of genes related to ROS scavenging were evaluated in WT and *PtOSH1*-OE*s*. A microplate reader (Bio-Rad, Hercules, CA, USA) was used to analyze the CAT, SOD, and POD activities before and after treatment with 60 μM CdCl_2_ based on the protocol of the Nanjing Jiancheng Bioengineering Institute.

### 2.4. Polymerse Chain Reaction (PCR) and Quantitative Reverse Transcription-PCR

PCR was performed as follows: 95 °C for 5 min; 35 cycles of 95 °C for 30 s, 58 °C for 40 s, and 72 °C for 30 s; and 72 °C for 10 min. SYBR Green Mix (Roche, Basel, Switzerland) was added to the PCR mixture, and quantitative reverse transcription-PCR (qRT-PCR) was conducted as follows: initial incubation at 95 °C for 5 min; 40 cycles of 95 °C for 30 s and 60 °C for 30 s; and 72 °C for 30 s. β-*Actin* of *P. trichocarpa* and *A. thaliana* were used as internal controls. Three independent biological experiments were analyzed with three technical repeats. The primers used are shown in the Appendix A.

## 3. Results

### 3.1. Molecular Characterization of a Poplar OSH1

We cloned an *OSH1* from *P. trichocarpa*. The open reading frame (ORF) of *PtOSH1* (POPTR_0016s12980) contained 813 nucleotides and encoded 270 amino acids. The predicted signal peptide of PtOSH1 was ‘MGSSPLLSFLVLTSLVVLFVTPSHS’ and located at the N-terminus of PtOSH1, this petide was necessary for its transportation to the lysosome. (Appendix A). Also, PtOSH1 was predicted to be located in the lysosome. The amino acid sequence of PtOSH1 had two characteristic motifs (CXXC and CQHGX_2_ECX_2_NX_4_C), and two glycosylation sites (NNT and NTS) (Appendix A). We speculated that PtOSH1 had sulfhydryl reductase activity. The homology of PtOSH1 with GILTs from other species was analyzed using Clustal Omega and BOXSHADE software. All of the GILTs had the characteristic CXXC and CQHGX_2_ECX_2_NX_4_C motifs and ten highly conserved cysteine residues (Appendix A), as well as sulfhydryl reductase activity, suggesting that PtOSH1 also has sulfhydryl reductase activity.

OsAK071633, OsAK106050, AT1G07080, AT4G12870, At4G12890, AT4G12900, AT4G12960, AT5G01580, CsXP_010475460, EgXP_010937929, NtXP_016471791, POPTR_0016s12980 were included in a phylogenetic tree constructed using MEGA 5.0 software. The OsAK071633, OsAK106050, AT1G07080, CsXP_010475460, EgXP_010937929, NtXP_016471791, POPTR_0016s12980 formed a large branch, suggesting close relationships among OsAK071633, OsAK106050, AT1G07080, CsXP_010475460, EgXP_010937929, NtXP_016471791, POPTR_0016s12980 (Figure 1). In addition, the tertiary structures of HoGILT, AtOSH1 and PtOSH1 were predicted using the online software tools of the SWISS-MODEL server (http://www.expasy.org/swissmod/SWISS-MODEL.html). Three-dimensional models are constructed according to multiple-threading alignments by LOMETS (https://zhanglab.ccmb.med.umich.edu/LOMETS/). A homology analysis showed that POPTR_0016s12980 differed from the HoNP_006323 and AT1G07080, but had a similar three-dimensional (3D) structure. The CXXC and CQHGX_2_ECX_2_NX_4_C motifs were present in the same position in the 3D structure of POPTR_0016s12980, AT1G07080 and HoGILT (Figure 2). Therefore, POPTR_0016s12980 may have similar biological functions to HoNP_006323.

### 3.2. Transcript Levels of PtOSH1 in Tissues and Under Cd^2+^ Stress

The highest transcript level of *PtOSH1* was in young and mature leaves, followed by roots, and the lowest was in petioles (Figure 3A). Treatment with 2 mM CdCl_2_ significantly increased the transcript level of *PtOSH1* from 6 to 48 h, with a peak at 12 h (Figure 3B). Treatment with 200 μM ABA caused significant accumulation of *PtOSH1* mRNA from 6 to 24 h (Figure 3C). The expression of *PtOSH1* was enhanced from 1 to 24 h of 2 mM H_2_O_2_ treatment (Figure 3D). Plant roots and leaves absorb exogenous substances. During plant growth and development, nutrient transport involves various signaling pathways, in which oxidoreductases play an important role. Therefore, the differential expression of GILT in different tissues may be related to normal physiological functions.

### 3.3. Expression, Purification, and Functional Analysis of Recombinant PtOSH1

The ORF of *PtOSH1* was cloned into the plasmid PET-28a between the *EcoR*I and *Xho*I sites. Analysis by 12% SDS-PAGE showed that the production of recombinant PtOSH1 was induced by 1 mM IPTG at 110 rpm at 16 °C, but not at 37 °C (Figure 4A). Recombinant PtOSH1 was purified using Ni-IDA resin (Figure 4B) and Western blotting showed that recombinant PtOSH1 was specifically recognized by rabbit antiserum (Figure 4B).

Arunachalam et al. [3] showed that GILT proteins can catalyze the reduction of disulfide bonds and exhibit sulfhydryl reductase activity under acidic conditions. We used human IgG as a substrate to assay the activity with PtOSH1 in vitro. The disulfide bond of IgG was intact (lane 2) following treatment of DTT (Figure 4C). In the absence of 25 mM DTT, the disulfide bond (lane 6) of IgG was intact at pH 4.5 and 7.0. At 10 mM DTT, the disulfide bond (lane 5) was broken at pH 7.0 but not pH 4.5. Activated PtOSH1 cleaved IgG into heavy and light chains (lane 3), indicating that recombinant PtOSH1 catalyzed the reduction of the disulfide bond, in which DTT at a low concentration acted as a cofactor.

### 3.4. Characterization of Transgenic A. Thaliana Lines

To further study the function of PtOSH1 in transgenic *A. thaliana* lines, the PBI121-*PtOSH1* vector was introduced into WT *A. thaliana*. Subsequently, *PtOSH1*-OE lines were obtained (Appendix A). The total genome of the WT and *PtOSH1*-OE lines was extracted and amplified by PCR. The lanes of the *PtOSH1*-OE lines, but not that of the WT, had the expected band (Appendix A). Also, qRT-PCR analysis showed that *PtOSH1* was stably expressed in *A. thaliana* (Appendix A). In addition, the transcript levels of Arabidopsis homologs At1G07080 and At5G01580 were identified. The expression of At1G07080 in *PtOSH1*-OE lines was increased significantly (*p* < 0.05) (Appendix A). However, there are no significant difference in expression of At5G01580 between WT and *PtOSH1*-OE lines (Appendix A).

### 3.5. Response of Transgenic Plants to Cd^2+^ Stress

WT and *PtOSH1*-OE *A. thaliana* plants were grown in 1/2 MS medium containing 20, 40, 60, and 80 μM CdCl_2_ and their growth was evaluated after 20 days. The growth of WT and transgenic *A. thaliana* lines was little affected by 20 μM CdCl_2_ (Figure 5A,B and Figure 6). When exposed to 40–60 μM CdCl_2_, the root length and biomass of plants decreased significantly, and the root length and biomass of the *PtOSH1*-OE *A. thaliana* lines were significantly higher than those of WT *A. thaliana* (Figure 5C,D and Figure 6), suggesting that transgenic plants have a growth advantage. At 80 μM CdCl_2_, the germination and growth of *A. thaliana* were inhibited; leaf yellowing, root length, and the fresh weight of the *PtOSH1*-OE *A. thaliana* lines were significantly higher than those of WT *A. thaliana*; and the number of lateral roots increased significantly (Figure 5E and Figure 6). Therefore, *PtOSH1*-*like* transgenic *A. thaliana* plants showed greater tolerance to CdCl_2_ stress than did WT plants.

Cd stress triggers the production of ROS in plants, the accumulation of which hinders photosynthesis. In addition, the metalloproteins related to electron transfer in plants are compromised by Cd, disrupting respiration [12]. The plant antioxidant defense system eliminates ROS to prevent the damage caused by oxidative stress [40]. qRT-PCR analysis showed that the expression levels of ascorbate peroxidase (*APX*), *CAT*, glutathione S-transferase (*GST*), *POD*, and *SOD* in WT and *PtOSH1*-OE plants were not significantly different under normal conditions but were higher in *PtOSH1*-OE plants than in WT plants under CdCl_2_ stress (Figure 7). To determine whether overexpression of *PtOSH1* increased tolerance to CdCl_2_ stress, we analyzed the activities of POD, SOD, and CAT under 60 μM CdCl_2_ stress. Under normal conditions, *PtOSH1*-OE *A. thaliana* lines had slightly higher activities of POD, SOD, and CAT than WT *A. thaliana*; however, *PtOSH1*-OE *A. thaliana* lines had significantly higher activities of POD, SOD, and CAT than WT *A. thaliana* after 20 days of CdCl_2_ stress (Figure 8). Therefore, PtOSH1 promotes ROS scavenging by POD, SOD, and CAT to alleviate the oxidative damage to membranes caused by CdCl_2_ stress, increasing the tolerance to CdCl_2_ of the *PtOSH1*-OE plants.

## 4. Discussion

Research on GILT has focused on animals, including human [1], mouse [41], chicken [42], zebrafish [43], amphioxus [44], pig [45], large yellow croaker [46], and fruit bat [47]. These GILTs have the CXXC/S functional domain, which is similar to the WCGH/PCK domain of the thioredoxin family [41,45]. In addition, GILT has a signature sequence (CQHGX_2_CX_2_NX_4_C), multiple N-glycosylation sites, and 10–11 conserved cysteine residues [48]. However, the function of plant GILTs was unknown. In this study, we characterized *PtOSH1* of *P. trichocarpa.* PtOSH1 has the same motifs and a similar structure to human GILT, indicating that these proteins have similar functions.

During the processing and presentation of antigens, disulfide bonds are denatured, renatured, and reduced; the latter is particularly important. GILT is the only reductase that catalyzes the reduction of disulfide bonds at low pH and exhibits high activity in the acidic lysosome [49,50,51]. Ohkama-Ohtsu et al. [37] demonstrated that recombinant At5g01580 protein expressed by *E. coli* had thiol reductase activity under neutral conditions. However, the GILT of humans has the highest thiol reductase activity at acidic conditions [3]. In this study, we demonstrated that PtOSH1 cleaves IgG into heavy and light chains by catalyzing the reduction of disulfide bonds under neutral conditions. PtOSH1-catalyzed reduction of disulfide bonds may alter the structure and function of proteins in poplar. The reduction of disulfide bonds in some proteins restores their normal physiological function. Therefore, PtOSH1 may affect homeostasis of poplar by catalyzing the reduction of disulfide bonds.

Plants under stress conditions produce large amounts of ROS, including superoxide anions and free radicals [52,53]. To decrease the resulting oxidative damage, the antioxidant systems of plants not only scavenge ROS and limit their production but also repair oxidative damage [54,55]. Thioredoxin reductase (Trx) has a WCGH/PCK domain, which catalyzes the reduction of disulfide bonds and is involved in a variety of biochemical reactions [56], including regulation of redox potential, antioxidants, signal-transduction pathways, transcription factors, and the response to heavy metal stress [30,57,58]. In this study, the expression of *PtOSH1* was upregulated by CdCl_2_ treatment. Also, overexpression of *PtOSH1* enhanced the resistance of *A. thaliana* to CdCl_2_ stress. These findings implicate *PtOSH1* in the response to CdCl_2_ stress. Trx scavenges ROS and activates proteins inactivated by oxidative stress, which are important for maintaining the physiological function of cells under oxidative stress [59,60,61]. In WT and *PtOSH1*-OE plants, *APX*, *CAT*, *GST*, *POD*, and *SOD* transcript levels were increased by CdCl_2_ stress, and the magnitude of the increase was higher in *PtOSH1*-OE plants than in WT plants. In addition, the activities of SOD, POD, and CAT were higher in *PtOSH1*-OE *A. thaliana* lines than in WT *A. thaliana*. Therefore, PtOSH1 might be important for defense against CdCl_2_ stress. However, the underlying mechanism is unknown; therefore, a mechanistic investigation of the antioxidant activity of PtOSH1 is required to determine its effect on ROS.

Previous studies showed that Cd poisoning can lead to oxidative stress and protein denaturation in plants. However, plants have defense mechanisms to alleviate the damage from oxidative stress, including increasing the ability to remove oxidized proteins, improving the synthesis of antioxidant molecules and molecular chaperones, and changing the composition of plant cell walls and xylem sediments [17]. In addition, Salt et al. [62] found that Cd was mainly distributed in the leaf epidermis and epidermal hairs of mustard. Cd in plants may be a defense mechanism to prevent damage because the substances needed for photosynthesis, growth, and development are mainly in mesophyll cells, and leaf epidermis and epidermal hairs play a role in isolation and protection. Due to the lack of GILT, the activity and stability of SOD2 were decreased in the animal cells. Recombinant GILT improved the activity and stability of SOD2 and maintained a relatively steady level of ROS; thus, GILT promotes maintenance of the redox state in cells. GILT also promotes intracellular oxidative stress in cells, which can accelerate autophagy and decompose damaged mitochondria [9]. In this study, the transcript level of *PtOSH1* was improved under Cd treatment, and PtOSH1 catalyzed the reduction of disulfide bonds. Two cysteines are located in active sites of GILT. One cysteine is nucleophilic; it attacks the disulfide bond on the substrate to form a disulfide compound and intermediate substrate, and then the two cysteines on GILT carry out an internal attack to cause the substrate to escape so that the disulfide bond of the substrate is opened [51]. GILT catalyzes disulfide reduction, which is accompanied by changes in protein structure and function in cells. Some disulfide bonds can be reduced to restore their normal physiological functions. GILT regulates ROS homeostasis by repairing antioxidant enzymes and ultimately maintains the redox state in cells. Together, our results demonstrate that PtOSH1 is essential for Cd stress and that PtOSH1 may restore antioxidant enzymes by catalyzing disulfide reduction as well as regulating the steady state of ROS.

## 5. Conclusions

In conclusion, when plants are stressed by heavy metals, the activity of the antioxidant system in plants is improved, and antioxidant enzymes can scavenge ROS and play a protective role. When stressed by heavy metals for a long time, the antioxidant enzyme system of plants is destroyed, and plants are poisoned by ROS. In addition, although GILT from animals has been well characterized and its function evaluated, there have been fewer studies of the function of GILT-like in plants. Therefore, a more thorough understanding of GILT, including how it is regulated, is important for maintaining the mechanism of ROS underlying Cd treatment. In this study, we isolated *PtOSH1* from *P. trichocarpa*, which can respond to Cd treatment; transgenic experiments in *A. thaliana* provided further evidence related to the response to Cd stress. Collectively, our results demonstrate that PtOSH1 catalyzes the reduction of disulfide bonds and may repair the antioxidant enzymes resulting from Cd stress and regulate the ROS-scavenging system, which is important for the steady state of plant cells.

## Figures and Tables

**Figure 1 cells-09-00076-f001:**
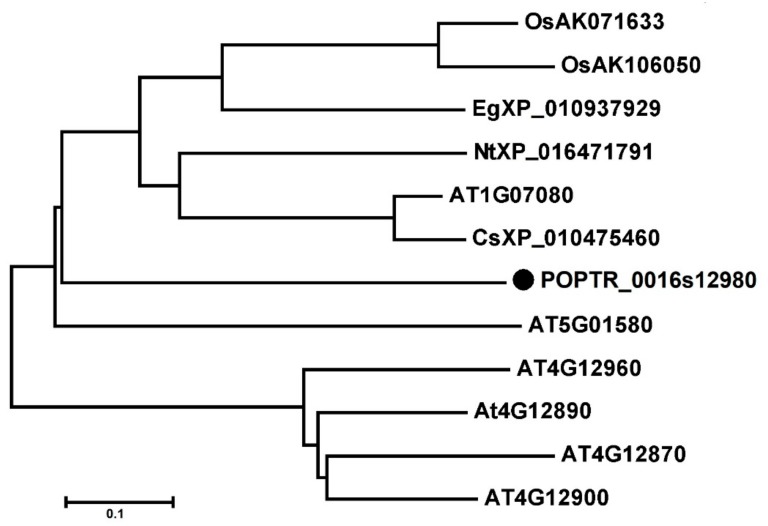
The phylogenetic tree analysis of OSH1 proteins. Translated amino acid sequence of POPTR_0016s12980 was compared using the BLASTP search program. GenBank accession number is preceded by a species identifier. Pt, *Populus trichocarpa*; At, *Arabidopsis thaliana*; Os, *Oryza sativa*; Nt, *Nicotiana tabacum*; Cs, *Camelina sativa*; Eg, *Elaeis guineensis*.

**Figure 2 cells-09-00076-f002:**
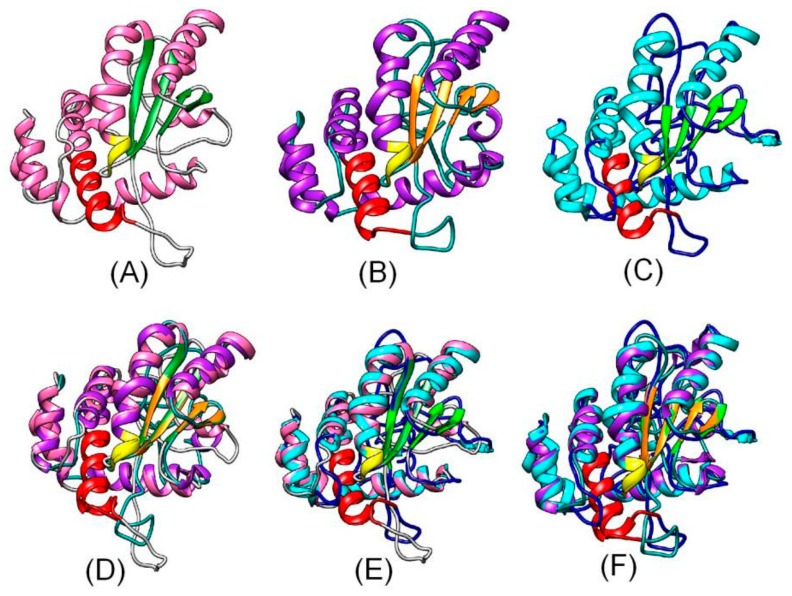
Homology of the predicted tertiary structures of HoNP_006323 (**A**) *H. sapiens* (NP_006323.2), and homology of the predicted tertiary structures of AT1G07080 (**B**), POPTR_0016s12980 (**C**), HoNP_006323.2 and AT1G07080 (**D**), HoNP_006323 and POPTR_0016s12980 (**E**), and POPTR_0016s12980 and AT1G07080 (**F**). Red, CXXC motif; yellow, CQHGX_2_ECX_2_NX_4_C motif. Pt, *P*. *trichocarpa*; At, *A*. *thaliana*; Ho, *Homo sapiens*.

**Figure 3 cells-09-00076-f003:**
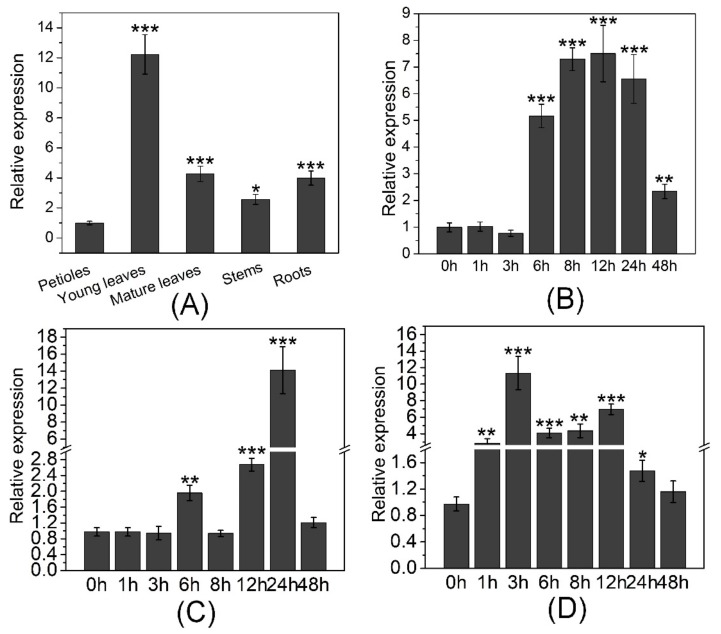
The identification of *PtOSH1* mRNA expression level based on the quantitative reverse transcription polymerase chain reaction (qRT-PCR). (**A**) The mRNA transcript levels of *PtOSH1* in various tissues. (**B**) The mRNA transcript level of *PtOSH1* in response to Cd stress. (**C**) The mRNA transcript level of *PtOSH1* in response to ABA stress. (**D**) The mRNA transcript level of *PtOSH1* in response to H_2_O_2_ stress. Three independent biological experiments were performed with three technical repeats. One-way analysis of variance (ANOVA) and Tukey’s test were used to evaluate significant differences. Data are 2^–ΔΔCt^ levels relative to the petiole and normalized to that of *PtActin*. Vertical bars represent the means ± standard deviation (SD, n = 3). *, Significant difference at *p* < 0.05; **, Significant difference at *p* < 0.01; ***, significant difference at *p* < 0.001.

**Figure 4 cells-09-00076-f004:**
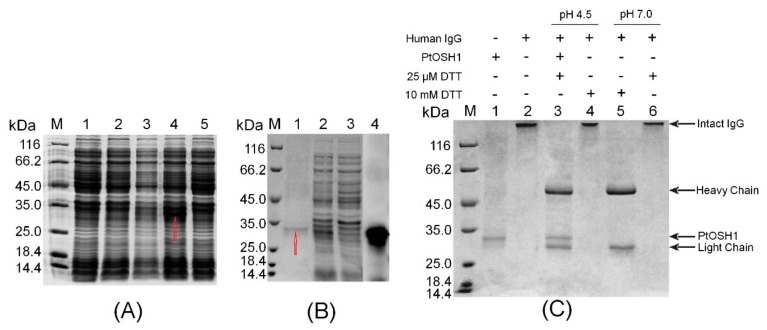
The identification of sulfhydryl reductase activity in vitro. (**A**) Analysis of recombinant PtOSH1 protein expression. Lane M, molecular mass marker; lane 1, control; lane 2, recombinant *E. coli BL21* (*DE3*) incubated at 220 rpm at 37 °C with induction by 1 mM isopropyl β-D-1-thiogalactopyranoside (IPTG); lane 3, recombinant *E. coli BL21* (*DE3*) incubated at 110 rpm at 37 °C with induction by 1 mM IPTG; lane 4, recombinant *E. coli BL21* (*DE3*) incubated at 110 rpm at 16 °C with induction by 1 mM IPTG; and lane 5, recombinant *E. coli BL21* (*DE3*) incubated at 220 rpm at 37 °C with induction by 1 mM IPTG. (**B**) Purification and western blot analysis of recombinant PtOSH1 protein. Lane M, molecular mass marker; lane 1, eluate; lanes 2–3, flow-through; lane 4, Western blot. (**C**) Analysis of PtOSH1 protein activity in vitro. Lane M, molecular mass marker; lane 1, recombinant PtOSH1 protein; lane 2, human IgG; lane 3, human IgG incubated with PtOSH1 protein (pH 4.5); lane 4, human IgG incubated with 10 mM dithiothreitol (DTT, pH 4.5); lane 5, human IgG incubated with 10 mM DTT (pH 7.0); and lane 6, human IgG incubated with 25 μM DTT (pH 7.0).

**Figure 5 cells-09-00076-f005:**
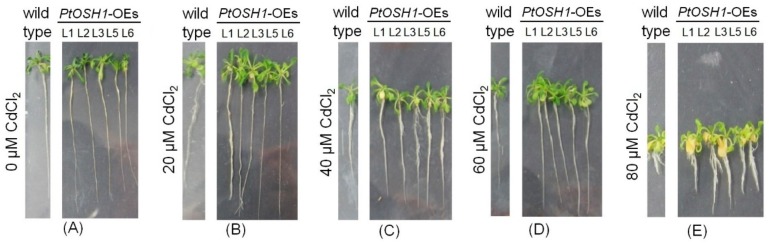
Phenotypic analysis of wild-type (WT) and transgenic *A. thaliana* lines (Line1 (L1), L2, L3, L5, L6) treated in 0 μM Cd (**A**), 20 μM Cd (**B**), 40 μM Cd (**C**), 60 μM Cd (**D**), and 80 μM Cd (**E**). Three independent biological experiments were performed.

**Figure 6 cells-09-00076-f006:**
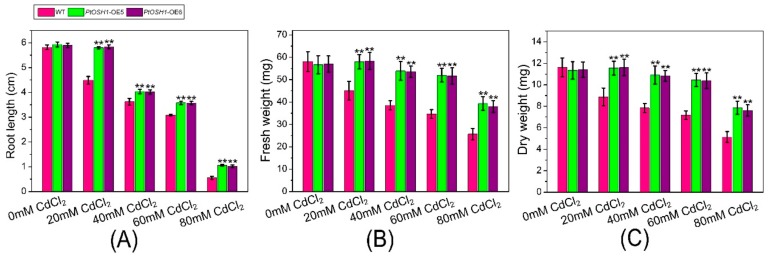
Tolerance of WT *A. thaliana* and *PtOSH1*-OE *A. thaliana* lines to Cd stress. (**A**) Root length of WT and *PtOSH1*-OE *A. thaliana* lines under Cd stress. (**B**) Fresh weight of WT and *PtOSH1*-OE *A. thaliana* lines under Cd stress. (**C**) Dry weight of WT and *PtOSH1*-OE *A. thaliana* lines under Cd stress. Three independent biological experiments were performed. One-way ANOVA and Tukey’s test were used to evaluate significant differences. Vertical bars represent the means ± SD (n = 3). **, Significant difference at *p* < 0.01.

**Figure 7 cells-09-00076-f007:**
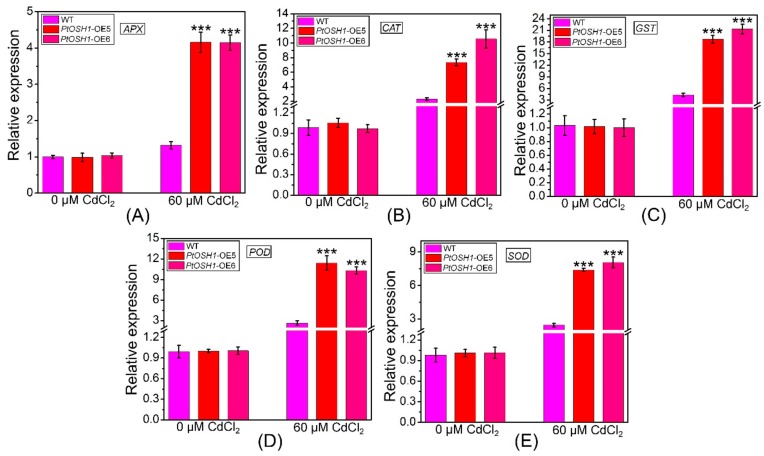
Transcript levels of ascorbate peroxidase (*APX*) (**A**), catalase (*CAT*) (**B**), glutathione S-transferase (*GST*) (**C**), peroxidase (*POD*) (**D**), and superoxide dismutase (*SOD*) (**E**) before and after exposure to 60 μM Cd^2+^. Data are 2^–ΔΔCt^ levels relative to WT plants and normalized to the transcript level of *AtActin*. Three independent biological experiments were performed with three technical repeats. One-way ANOVA and Tukey’s test were used to evaluate significant differences. Vertical bars represent the means ± SD (n = 3). ***, Significant difference at *p* < 0.001. Accession numbers: *APX* (NM_119666.4), *CAT* (NM_119675.4), *GST* (NM_128499.5), *POD* (NM_119917.3), and *SOD* (NM_001084025.1).

**Figure 8 cells-09-00076-f008:**
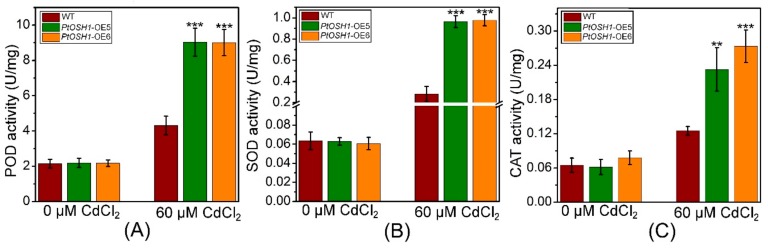
Activities of POD (**A**), SOD (**B**), and CAT (**C**) in WT and *PtOSH1*-OE *A. thaliana* lines before and after exposure to 60 μM Cd^2+^. Three independent biological experiments were performed with three technical repeats. One-way ANOVA and Tukey’s test were used to evaluate significant differences. Vertical bars represent the means ± SD (*n* = 3). **, Significant difference at *p* < 0.01; ***, significant difference at *p* < 0.001.

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
