# Peer review of "Identification and Characterization of an OSH1 Thiol Reductase from Populus trichocarpa"

_cells, 2019, doi:10.3390/cells9010076_

Round 1
Reviewer 1 Report
The manuscript by Wei et al. describes the characterization of an interferon-inducible lysosomal thiol reductase from poplar. While the manuscript is generally well written, I have major concerns about the experimental setup and statistics. For example, results in Figure 3b, c, and d show increased expression of GILT in poplar, but no information is given about what leaf was used for the study, while in Figure 3a large differences between different organs are shown. The authors should describe which leaves were used for the follow-up experiments. Also, in this figure as well as in some of the following a ttest was supposedly used, but that is not described anywhere in the text. For most of these results an ANOVA should at least be used since multiple experiments are compared. In the methods section it says that three independent biological experiments were performed. It is however unclear what this means. Were 3 experiments performed with one biological replicate each or were 3 biological replicated done? This is particularly strange for the results in Figure 5, but also others and needs clarification. Figure 7 is too small to be readable and should be increased. Some sentences or statements should be altered. For example, line 132 "the first amino acid of which was methionine". Methionine is always the first amino acid and does not need to be explicitly mentioned. Line 181, it should read "was cloned" rather than "ligated". Line 322, the first sentence is a statement that looks out of place. It has been known for quite some time that plants have an anti-oxidant system. This is now new and not based on the results presented in this paper. Thiol reducates play an important role in redox regulation and should be discussed in this context. The phylogenetic tree is somewhat confusing since it places plant between invertebrates and all other animals.
These points need clarification.
Author Response
Review 1Comments and Suggestions for Authors
The manuscript by Wei et al. describes the characterization of an interferon-inducible lysosomal thiol reductase from poplar. While the manuscript is generally well written, I have major concerns about the experimental setup and statistics.
Dear Reviewer:
Thank you for the reviewers’ comments concerning our manuscript. Thanks very much for spending a lot of time and energy reviewing our manuscript. Those comments are all valuable and very helpful for revising and improving our paper, as well as the important guiding significance to our researches. We are pleased to note the favorable comments of reviewers in their opening sentence.
Thank you and best regards.
Yours sincerely,
For example, results in Figure 3b, c, and d show increased expression of GILT in poplar, but no information is given about what leaf was used for the study, while in Figure 3a large differences between different organs are shown.
Answer: According to reviewer’s comment, we modified the Figure 3 and added the information about the leaf materials for study,
The authors should describe which leaves were used for the follow-up experiments. Also, in this figure as well as in some of the following a ttest was supposedly used, but that is not described anywhere in the text. For most of these results an ANOVA should at least be used since multiple experiments are compared. In the methods section it says that three independent biological experiments were performed. It is however unclear what this means. Were 3 experiments performed with one biological replicate each or were 3 biological replicated done? This is particularly strange for the results in Figure 5, but also others and needs clarification.
Answer: Thanks for the valuable comment. In this study, three independent biological experiments were performed with three technical repeats. One-way ANOVA and Tukey’s test were used to evaluate significant differences.
Figure 7 is too small to be readable and should be increased.
Answer: Thanks for reviewer’s suggestion. We have modified the Figure 7 according to reviewer’s comment
Some sentences or statements should be altered. For example, line 132 "the first amino acid of which was methionine". Methionine is always the first amino acid and does not need to be explicitly mentioned. Line 181, it should read "was cloned" rather than "ligated". Line 322, the first sentence is a statement that looks out of place. It has been known for quite some time that plants have an anti-oxidant system. This is now new and not based on the results presented in this paper. Thiol reducates play an important role in redox regulation and should be discussed in this context. The phylogenetic tree is somewhat confusing since it places plant between invertebrates and all other animals.
Answer: We modified the sentences mentioned above according to the reviewer's suggestion.
Reviewer 2 Report
The manuscript by Wei et al., reports the identification of interferon in Populus trichocarpa. Based on the presented data, the plant interferon is structurally similar to the human interferon. Indeed, the purified PtGILT shows activity when human IgG is used as a substrate. Additionally, the authors show some data indicating that PtGILT plays some role in the plant adaptation to Cd toxicity.
The data presented in this manuscript is interesting, indeed. However, the way that the authors show this interesting data obscure their relevance. The authors need to address some questions before considering this manuscript further
My main concern with this manuscript is that the authors do not provide an appropriate background. This makes very difficult to understand the rationality of this study. For instance, what is known about interferons in plants? is there any relationship between interferons and plant immunity? It is not clear the link between Cd stress and interferon. Why Poplar? All these issues must be addressed
131-132: Please clarify, what kind of signal peptide?
It is very important that authors explain to Populus how the quaternary structure was obtained?
Again, the authors must explain the rationality of this study, why do they decide to analyze the expression under Cd stress and in response to hormones?
Since the authors decided to work on Arabidopsis, it is important that they identify Knockout mutants, analyze the phenotype, and complement the mutants with either AtGILT or PtGILT
It is important to show that the authors assess the ROS accumulation in both Wt, Ox, knockout mutant.
Round 2
Reviewer 2 Report
The authors have addressed all my comments
Author Response
Review 2 Comments and Suggestions for Authors The manuscript by Wei et al., reports the identification of interferon in Populus trichocarpa. Based on the presented data, the plant interferon is structurally similar to the human interferon. Indeed, the purified PtGILT shows activity when human IgG is used as a substrate. Additionally, the authors show some data indicating that PtGILT plays some role in the plant adaptation to Cd toxicity. The data presented in this manuscript is interesting, indeed. Dear Reviewer: Thank you very much for spending a lot of time and energy reviewing our manuscript. Also, thanks for your patience in pointing out the mistakes made by our poor knowledge. Those comments are all valuable and very helpful for revising and improving our paper, as well as the important guiding significance to our researches. We have studied comments carefully and have made correction which we hope meet with approval. Thank you and best regards. Yours sincerely, Hui Wei However, the way that the authors show this interesting data obscure their relevance. The authors need to address some questions before considering this manuscript further. My main concern with this manuscript is that the authors do not provide an appropriate background. This makes very difficult to understand the rationality of this study. For instance, what is known about interferons in plants? is there any relationship between interferons and plant immunity? It is not clear the link between Cd stress and interferon. Why Poplar? All these issues must be addressed Answer: Thanks for the valuable comments. We added the background. In our lab, we also used poplar as a model plant to analyze the function of genes from poplar and provided a theoretical basis for breeding work. Generally speaking, the identification of Arabidopsis functional genes will provide some theoretical support for us to study the genes of other species. However, there is no relevant report on GILT of plants. In this study, we selected GILT from poplar for functional identification, hoping to provide some theoretical basis for the study of GILT. 131-132: Please clarify, what kind of signal peptide? Answer: Thanks for the valuable comments. In this study the predicted signal peptide of PtGILT was ‘MGSSPLLSFLVLTSLVVLFVTPSHS’, and the first amino acid of signal peptide was methionine located at the N-terminus of PtGILT which was necessary for its transportation to lysosomal system. It is very important that authors explain to Populus how the quaternary structure was obtained? Answer: Sorry for Fig.9 which was unclear to explain the structure of GILT protein of Populus. The new figure is to replace Fig.9. and the GILT protein of Populus is only one subunit, the tertiary structures of which were predicted using the online software tools of the SWISS-MODEL server (http://www.expasy.org/swissmod/SWISS-MODEL.html). Again, the authors must explain the rationality of this study, why do they decide to analyze the expression under Cd stress and in response to hormones? Answer: Thanks for the good question. Although the function of GILT has not been identified in plants, thioredoxin (Trx), which has a similar structure as GILT, has been widely studied (Lillig et al., 2007; Meyer et al., 2002; Collin et al., 2003). In redox reactions, Trx in its reduced state contacts the disulfide bond of the target protein through the hydrogen on the sulfhydryl, and transforms its own sulfhydryl into a disulfide bond. At the same time, the disulfide bond on the target protein is disrupted, completing the reduction of Trx (Holmgren et al., 1995, Miki et al., 2012); GILT-mediated reduction occurs in a similar manner. Trx can function as an antioxidant to remove ROS or as a repair enzyme to regenerate proteins inactivated by oxidative stress. These studies provide a new theoretical basis for characterization of GILT from plants. Therefore, we identified the transcript level of PtGILT under the Cd stress and the result showed that the PtGILT can respond to Cd stress. Moreover, In order to study whether the PtGILT is related to ROS in plants, we analyzed the expression level of PtGILT under the H2O2 stress. Reference: Lillig, C.H.; & Holmgren, A. Thioredoxin and related molecules–from biology to health and disease. Antioxidants & Redox Signaling. 2007, 9, 25-47. Meyer, Y.; Vignols, F.; & Reichheld, J.P. Classification of plant thioredoxins by sequence similarity and intron position. Methods in Enzymology. 2002, 347, 394. Collin, V.; Issakidis-Bourguet, E.; Marchand, C.; Hirasawa, M.; Lancelin, J.M.; Knaff, D.B.; & Miginiac-Maslow, M. The Arabidopsis plastidial thioredoxins New functions and new insights into specificity. Journal of Biological Chemistry. 2003, 278, 23747-23752. Holmgren, A. Thioredoxin structure and mechanism: conformational changes on oxidation of the active-site sulfhydryls to a disulfide. Structure. 1995, 3, 239-243. Miki, H.; & Funato, Y. Regulation of intracellular signalling through cysteine oxidation by reactive oxygen species. The Journal of Biochemistry. 2012, 151, 255-261. Since the authors decided to work on Arabidopsis, it is important that they identify Knockout mutants, analyze the phenotype, and complement the mutants with either AtGILT or PtGILT Answer: Thanks for the creative opinion. In this study we analyzed the phenotype of the Arabidopsis overexpressing with PtGILT. From a point of view, these results enlighten the possibility of improving Populus by manipulating in molecular breeding. As the reviewer pointed out, it is important to identify Knockout mutants. We also clone the AtGILT (NM_100582.4) from Arabidopsis thaliana. The open reading frame (ORF) of AtGILT contained 798 nucleotides and encoded 265 amino acids. Also, the activity of recombinant AtGILT showed that AtGILT can catalyze the reduction of disulfide bonds. However, because this experiment just started, we only completed gene cloning and detection of enzyme activity in vitro. Arabidopsis mutants have not been obtained. The phenotype analysis of the Knockout mutants with AtGILT will continue studying in our laboratory for a better understanding of GILT functional role in plants. (A) Gel electrophoresis analysis of AtGILT (NM_100582.4). (B) Purification and western blot analysis of recombinant AtGILT protein. Lane M, molecular mass marker; lane 1, eluate; lanes 2–3, flow-through; lane 4, Western blot. (C) Analysis of AtGILT protein activity in vitro. Lane M, molecular mass marker; lane 1, recombinant AtGILT protein; lane 2, human IgG; lane 3, human IgG incubated with PtGILT protein (pH 4.5); lane 4, human IgG incubated with 10 mM DTT (pH 4.5); lane 5, human IgG incubated with 10 mM DTT (pH 7.0); lane 6, human IgG incubated with 25 μM DTT (pH 7.0). It is important to show that the authors assess the ROS accumulation in both Wt, Ox, knockout mutant. Answer: Thanks for the valuable suggestion. The ROS accumulation in Wt and only Ox mutant with PtGILT is important and provides the basis for functional role of PtGILT in plants. Because the analysis of AtGILT (NM_100582.4) is just going on. In the next experiment, we will design sgRNA and construct knockout plasmid of PRGEB31-SgRNA. Agrobacterium infection was performed and the mutant of Arabidopsis thaliana was obtained. Then, according to the valuable suggestions put forward by reviewer, a series of phenotypic and physiological and biochemical analysis was carried out in both WT, OE, knockout mutant. Also, further study in our laboratory will include analysis of the ROS accumulation in Wt and knockout mutant with PtGILT.
